# Macroeconomic Accounting of Water Resources: An Input-Output Approach to Linkage Analysis and Impact Indicators Applied to the State of Ceará, Brazil

**Rogério Barbosa Soares** [1,2,*], **Samiria Maria Oliveira da Silva** [1] , **Francisco de Assis de Souza Filho** [1] and **Witalo de Lima Paiva** [1]

1   Department of Hydraulic and Environmental Engineering, Federal University of Ceará, Fortaleza 60020-181, Brazil; samiriamaria@ufc.br (S.M.O.d.S.); assis@ufc.br (F.d.A.d.S.F.); witalo.paiva@ipece.ce.gov.br (W.d.L.P.)
2   Technical Institute of Research and Economic Strategy of Ceará, Fortaleza 60822-325, Brazil
*   Correspondence: rogeriosoares77@gmail.com; Tel.: +55-85-98775-4925

**Abstract:** This work aims to identify the key sectors of the economic structure, considering their water flows, and estimate each sector's impact. The goal is to highlight systemic characteristics in the regional economy, establish water use priorities, and assess water security. Based on a regional input-output matrix, we use the following methodologies: the Rasmussen and Hirschman indices for the 'forward and backward linkages'; simple multipliers of production, job, and income; and the elasticity of water consumption to final water demand. Thirty-two economic sectors and household consumption are analysed. From the elasticity of final water demand, we find that both trade and household consumption put more pressure on water consumption. Furthermore, a joint analysis of the applied methodologies shows that: (a) the trade sector is more relevant for the linkage of water flows, (b) the agriculture sector has the highest direct water consumption, and (c) the public administration sector has the highest intermediate water consumption.

**Keywords:** input-output matrix; water flows; regional economy



## 1. Introduction

Water resource allocation decisions among the production sectors have been widely discussed by educational and research institutions because they influence regional economic development [1–3], mainly in semiarid regions where water scarcity and water stress affect both the quantity and quality of available water [4].

Water scarcity and extreme weather events that accentuate its effects increase competition for water resources, and generate negative impacts both in the social and economic spheres, in addition to limiting the process of water allocation for the production of goods and services, and may even restrict the level of production of various economic sectors [5]. This is because most sectors are sensitive to water deficits during different stages of production.

In this sense, according to [6] the scarcity of water shapes the responsiveness of its users, and a deeper understanding is necessary of their interactions and the corresponding stimuli on economic sectors, via which the knowledge of sectoral and intersectoral water consumption within the regional productive structure can generate useful criteria for the planning and management of water resources in semiarid regions, making clearer the perception of its multiple uses.

Therefore, water resource allocation is associated with the management of supply, demand, and conflicts. These are relevant pillars in the decision-making process of water resource managers, including the analysis of regional support capacity that considers water availability and its multiple uses to avoid a situation of scarcity.

According to [7,8], regions with water scarcity problems should seek a better understanding of demand, knowing that water use is largely affected by the following factors:

economic development, the rapid urbanization process, changes in people's lifestyle and consumption, climate change, and population growth. The latter puts pressure on production systems by increasing the consumption of goods and services, which often use water resources intensively.

Therefore, recognizing the demand for water as a production factor among the various economic sectors will allow a better association between the forces of supply and demand in regions with problems of scarcity, in addition to improving the management of conflicts regarding its allocation, including the analysis of regional support capacity when considering the availability of water and its multiple uses to avoid a situation of economic loss caused by a water crisis. These are relevant pillars in the decision-making process of water resource managers [9,10].

In this sense, several studies have concentrated efforts on understanding water demand through the lens of economic activities. [11] evaluated the economic value of water storage and the consequences of this assessment for water management in the context of future droughts. The authors linked hydro-economic models to historical water management operations concerning their uses. An optimization model was presented by [12] for the allocation of water and agricultural resources under uncertainty. The authors analysed economic, environmental, and social considerations in an irrigated agricultural system. Ref. [13] performed a macroeconomic analysis of water use using an integrated model that combined both environmental engineering and economics. Ref. [14] built an integrated model combining a multi-objective optimization model with in-flow analysis to study the tradeoffs between economic growth, water use, and environmental protection.

The development of these hydro-economic models provided a better knowledge of the demand for water resources between the economic sectors and their interrelations within the productive structure, which according to [4,15] can reduce the uncertainties of forecasts associated with the volume of water allocated to production processes based on their direct and intermediate consumption, besides assisting in the future management of water resources in semiarid regions.

In this context, the use of the Input-Output Matrix (IOM) allows the identification and quantification of the direct and indirect use of water throughout the supply chain. Furthermore, the monetary value of water use can be expressed in the production of goods and services, and in economic transactions that occur in all sectors of an economy. This can help address the sectoral interdependencies of water use [16–18].

According to [19], the input-output analysis model provides a useful framework for tracking the use of a resource and its associates in the economic sectors and their interrelationships. For example, some studies [20–24] evaluated water demand of the economic sectors by applying the input-output analysis model to the regional economic structure.

Other applications of IOM models include the identification of water flows generated by interregional trade in goods and services between production and consumption centres (e.g., [25–27]). Furthermore, [28] examined water demand by evaluating the structural decomposition of the economy using the IOM together with network theory. Previous studies have also emphasised the joint analysis of water and energy consumption as factors in production in economic activities in urban areas using a multiregional in-flow model [29,30].

In these studies, the IOM model is associated with the concept of virtual water as an approach that aims to improve the level of water management, mainly in areas with water scarcity [31]. This concept investigates the volume of water incorporated in the production process of goods and services in a direct or intermediate way. The flow of virtual water occurs between the various economic sectors through supply chains accompanying the flow of trade in the local and regional markets [15,27,32]. This approach also allows expression of the monetary value of water use in the production of goods and services, enabling maximization of gross added value of production associated with the consumption of water resources, and the visualization of economic interactions that occur in all sectors of

an economy, in order to also allow the measurement of the marginal value of water for different users [33–37]

This study seeks to evaluate the economic structure of the State of Ceará, located in north-eastern Brazil, through the direct and indirect use of water resources among its economic sectors. Furthermore, we perform sectoral and intersectoral analyses of the economic impact of the consumption and exchange of water resources. Our aim is to better qualify the typology of water use and its economic efficiency to assist in the formulation of strategies by decision-makers. We apply the following methodologies: simple multipliers of production, employment, and income, Rasmussen and Hirschman indices, and the elasticity of water consumption with final water demand.

The simple multipliers of production, employment, and income estimate each economic sector's impact in terms of direct water consumption. This allows us to determine the most important relationships within the production process; that is, identifying sectors with better water use efficiency in their production processes, thereby measuring the impact of sectoral water consumption within the regional economic structure.

The Rasmussen and Hirschman indices on 'forward and backward linkages' reveal the sectors with the greatest power to propagate indirect water consumption within the economy through their demand, and the intersectoral supply of goods and services in the form of productive resources. This helps us observe direct, indirect, and induced effects.

Furthermore, the elasticity of resource consumption points to the sector with greater power to propagate the impact of demand shocks on the regional economic structure considering a 1% variation in final water demand. Our analysis differs from other studies as we consider the arrangement of the existing economic structure, and how it propagates the effects of external and internal shocks on final water demand.

Using these intersectoral analysis techniques developed through the IOM model allows us to estimate the level of impact of water consumption, directly and indirectly, in each of Ceará's productive sectors according to their systemic characteristics in the regional economy. Thus, these results allow us to identify the use of water in Ceará's economy and the establishment of priorities in decision-making about economic activities.

We propose that these decisions should be based on such a joint analysis. This is because water is a primary production resource, and its level of demand for economic activities can affect regional water security. Therefore, the formulation of public policies aimed at the allocation of water resources is important.

Ceará has increasingly evident water scarcity problems along with frequent droughts. This has generated disputes between the multiple users associated with agriculture, industry, urban, and residential consumption [38]. Our methodological techniques and procedures can be developed and replicated in other regions, states, or countries that suffer from the effects of water scarcity. This can help strengthen their information systems and integrate their decision-making processes on the management of water resources.

## 2. Materials and Methods

### 2.1. Study Area

The State of Ceará covers an area of 148,894.75 km$^2$ with a population of 9,187,103 in 2020. Its territorial boundaries spread from the Atlantic Ocean in the north, the State of Pernambuco in the south, the states of Rio Grande do Norte and Paraíba in the east, and the state of Piauí in the west (Figure 1). It has 184 municipalities, with 95% located in the semi-arid region of northeastern Brazil [39,40].

The state's economy constituted 2.25% of Brazil's gross domestic product (GDP) in 2017. It occupied the 11th position in the country and the 3rd in the northeast region. In nominal values, the state's GDP reached R$147,890 million in 2017 [41].

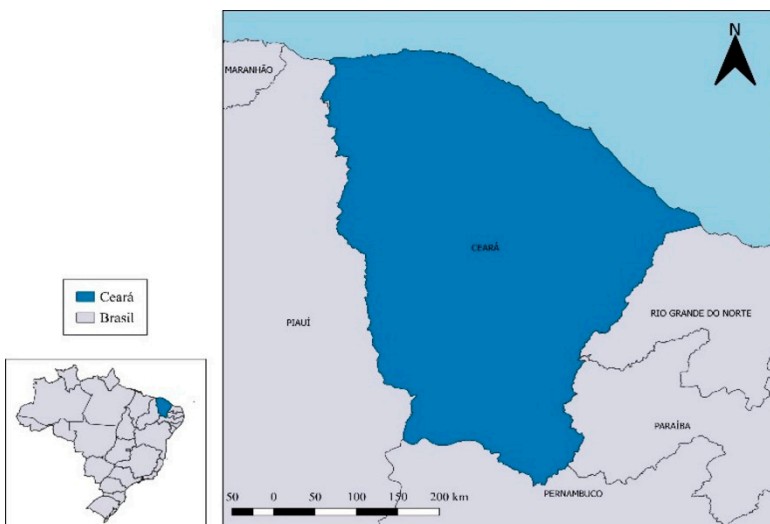

**Figure 1.** Location of the State of Ceará.

### 2.2. Methodology

Figure 2 outlines our methodology. The water flows of Ceará's economic sectors were obtained through the IOM. They were used to analyse the linkages and their effects on the regional economy, and identify the key sectors through the following analysis tools: (1) Rasmussen and Hirschman connection indices "backward and forward linkages", (2) elasticity of water resource demand due to variation in final demand, and the use of (3) simple multipliers of production, employment, and income, to evaluate water productivity among productive sectors.

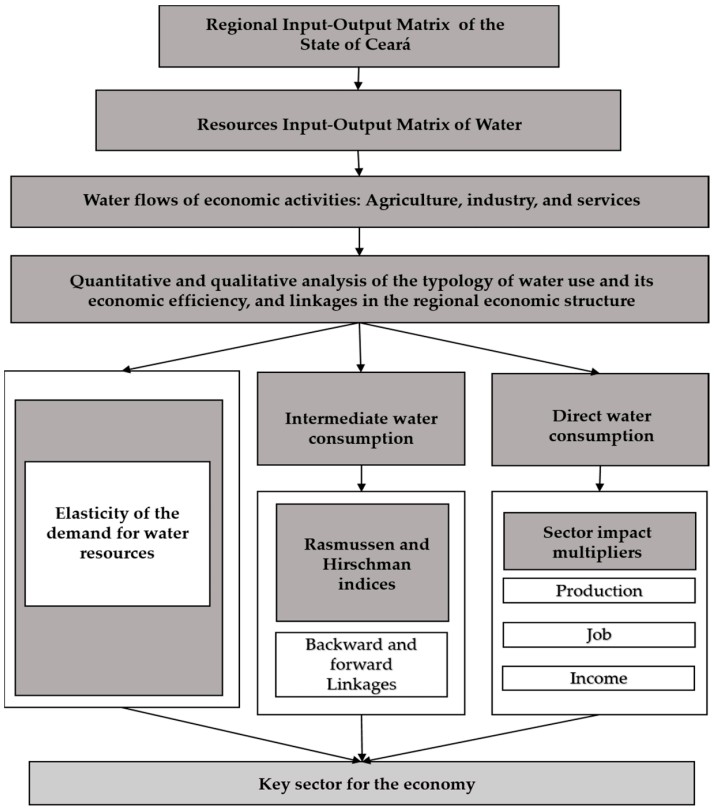

**Figure 2.** Methodological strategy.

Ceará's regional IOM (RIOM/CE) was built using the Institute of Research and Economic Strategy of Ceará's (IPECE) IOM model for the year 2013, considering 32 economic sectors, household consumption, and 58 products [42]. The IOM of Water Resources of Ceará (RIOM-WR) was also developed by [3] considering the same sectors and products as in RIOM/CE.

Household consumption refers to household demand for goods and services to meet immediate needs, including durable goods but not involving stock formation. Importantly, household consumption does not cover the purchase of residential or non-residential properties, nor the purchase of land [42]. Household consumption impacts the regional economy through the income acquired via their allocation of labour as entrepreneurs in the production of market and non-financial goods and services, and eventually in financial services [42].

### 2.2.1. Data Sources

The water consumption vector was constructed based on the application of technical coefficients of water consumption for each product listed in the RIOM-WR according to the typology of economic sectors belonging to agriculture, industry, or services (Figure 3).

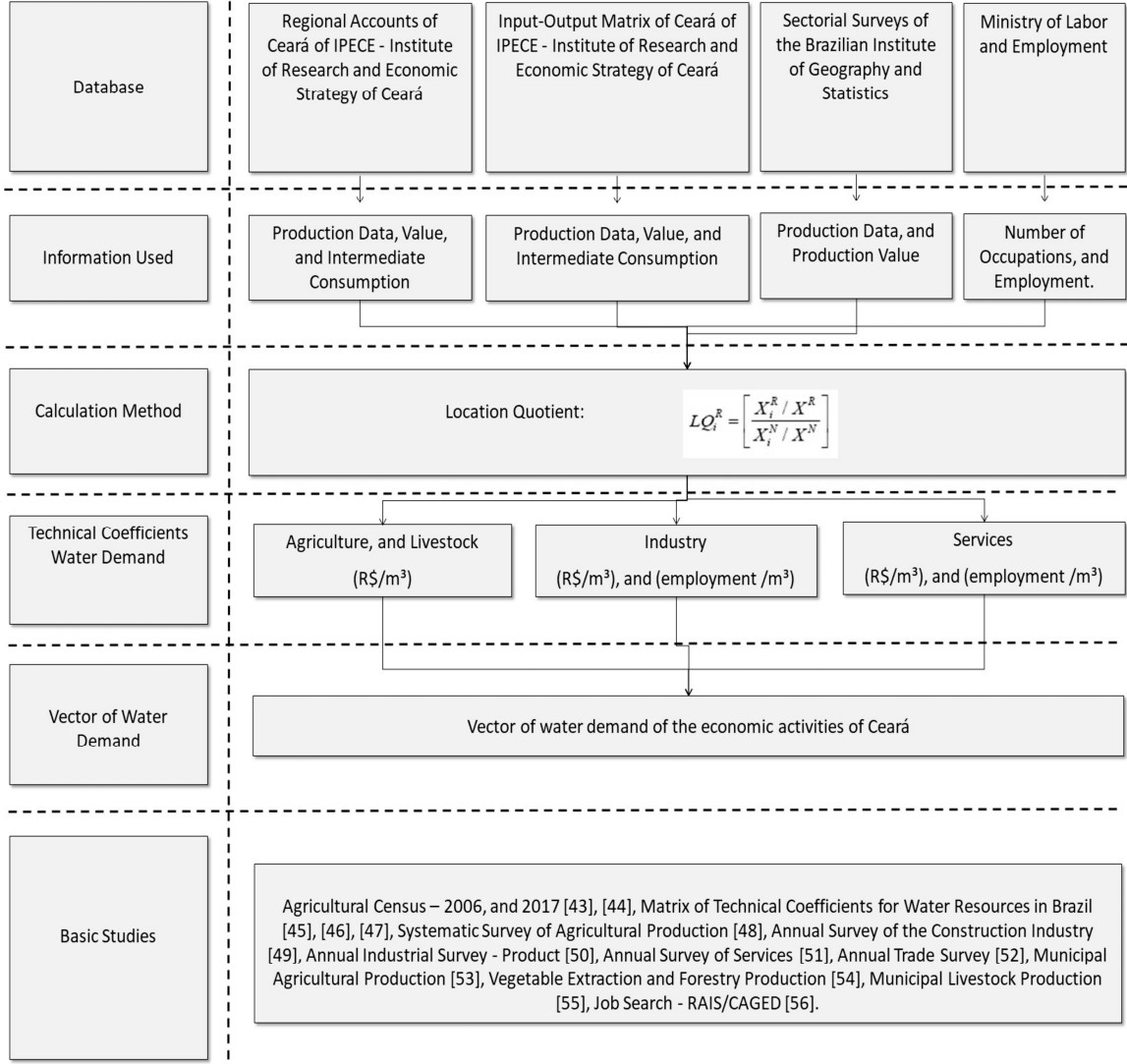

**Figure 3.** Methodology of the construction of technical coefficients and vector of water demand in the economic activities of Ceará [43–56].

After incorporating water supply, the RIOM is represented as observed in Table 1.

**Table 1.** Input-Output Matrix model after the incorporation of the water supply.

| | | Intermediate Consumption | | | Final Demand | Total |
|---|---|---|---|---|---|---|
| | | Consumer Sectors | | | | |
| | | Sector 1 | Sector 2 | Sector 3 | | |
| Productive Sectors | Sector 1 | Z11 | Z12 | Z13 | Y1 | Z1 |
| | Sector 2 | Z21 | Z22 | Z23 | Y2 | Z2 |
| | Sector 3 | Z31 | Z32 | Z33 | Y3 | Z3 |
| Water Supply | | Dw1 | Dw2 | Dw3 | Yw | D |

Source: [19,57].

2.2.2. Sectoral Impact Multipliers

Sector multiplier analysis is a traditional approach derived from the IOM and is one of the first analytical features provided by the IOM model.

Multipliers allow us to evaluate impacts on the economic system due to exogenous shocks. Three types of multipliers were used: (1) a production multiplier, which measures the effect on the production of all sectors of the economy; (2) an employment multiplier, which analyses the effect on the number of workers employed in all sectors of the economy; and (3) an income multiplier, which measures the effect on income from households [19,47], to evaluate water productivity among productive sectors.

(a)  Production Multiplier

The production multiplier for each sector is the sum of its respective column in Leontief's inverse matrix (B). The multiplier corresponds to the variation of the total production (direct and indirect) of the economy because of an exogenous unit variation of water ($1 \text{ m}^3$) in the final water demand of a given sector.

The multiplier is measured in terms of monetary units per water unit (R$ $1.00/\text{m}^3$ water) in the final water demand of a specific sector of the economy.

In formal terms, the simple product multiplier for sector j, $O_j$, is given by:

$$O_j = \sum_{i=1}^{n} b_{ij} \tag{1}$$

where, j is a certain sector of the economy, and $b_{ij}$ symbolises the elements of Leontief's inverse matrix.

(b) Job Multiplier

The employment multiplier estimates the direct and indirect effects of an exogenous change in final water demand on the employment generated in all economic sectors to meet the total production (direct and indirect), for example, of sector j in response to a $1 \text{ m}^3$ variation in the final water demand of the economy.

The coefficient of employment is represented by:

$$W_j = \frac{e_j}{X_j} \tag{2}$$

where $W_j$ is the coefficient of employment, $e_j$ corresponds to the number of workers employed in sector $j$, and $X_j$ is the gross value of production in sector $j$. For an economy with n sectors, one has:

$$w' = [w1, w2, \ldots, wn] \tag{3}$$

where $w'$ is a vector n × 1, whose elements are the employment coefficients of the n sectors of the economy. Now, $W$ is an array of the n × n order, whose main diagonal is given by

the elements of the vector $w'$ (Equation (3)). Furthermore, it has zeros outside the main diagonal. From $W$ (vector of the coefficient of employment) and B (Leontief inverse), one can create an array E of the same order as follows:

$$E = WB \tag{4}$$

Each element of $E$ is given by $e_{ij} = w_i \times b_{ij}$. This is interpreted as the amount of employment generated in sector $i$ to meet the total production (direct and indirect) of sector j in response to a 1 m³ variation in the final water demand of the economy.

Through matrix $E$, we obtained a sector's employment multiplier per 1000 m³ of water consumed by that sector (Job/1000 m³ water). Because the structure of matrix $E$ is similar to the structure of matrices B and A (matrix of technical coefficients), the simple multiplier of employment is given by:

$$E_j = \sum_{i=1}^{n} e_{ij} = \sum_{i=1}^{n} w_i b_{ij} \tag{5}$$

Similar to the calculation of the production multiplier, the employment multiplier of sector j is calculated as the sum of the elements of the $j^{th}$ column of matrix $E$. We similarly obtained each sector's multiplier.

(c) Income Multiplier

The income multiplier measures the impact of a unit variation in final water demand on household income. This is calculated using the relationship between the income (salary) generated in a given sector and the volume of water consumed in its production process. Formally, it can be represented as follows:

$$R_j = \frac{l_j}{X_j} \tag{6}$$

where $l_j$ is the income generated in sector $j$, and $X_j$ is the volume of water consumed by the sector's production activities.

Analogous to the employment multiplier, in an economy with n sectors, one can construct a vector n × 1 referring to the income generation coefficients, '$r$':

$$r = [r1, r2, \ldots, rn] \tag{7}$$

Now, $r$ is an array of n × n order, whose main diagonal is given by the elements of the $r$ vector and has zeros outside the main diagonal. From $R$ and $B$, one can create an array $MR$ of the same order as follows:

$$MR = RB \tag{8}$$

Each $MR$ element is given by $MR_{ij} = r_i \times b_{ij}$. This can be interpreted as the income generated in sector $i$ to meet the total production (direct and indirect) of sector '$j$' in response to a variation of R\$1.00 in sector $j$'s final water demand in terms of a 1 m³ variation in the final water demand of the economy.

Thus, the $MR$ matrix provides the sectoral structure of income generation in the economy, per additional unit of final water demand. As the structure of the $MR$ matrix is similar to the structure of matrices $B$ (Leontief inverse) and $A$ (matrix of technical coefficients), the simple income multiplier is given by:

$$MR_j = \sum_{i=1}^{n} MR_{ij} = \sum_{i=1}^{n} r_i b_{ij} \tag{9}$$

Similar to calculating the production and employment multipliers, the income multiplier of sector $j$ is the sum of the elements of the $j^{th}$ column of the $MR$ matrix.

### 2.2.3. Rasmussen and Hirschman Indices

The Rasmussen and Hirschmann indices [58,59] were applied to RIOM/CE to determine which sectors would show the highest forward and backward linkages within the economy. The 'backward' linkage indices reveal how much a sector would demand from the others, while the 'forward' linkage indices reveal how much other sectors would demand from the sector under consideration. That is, these indices indicate the degree to which a sector demands (supplies) inputs from (to) other sectors of the economy.

These variability measures help one to identify the interrelationship, or rather make it possible to verify the dispersal capacity in the other sectors due to the impact on one sector.

For computing these linkages, following [19,60], we used the elements of Leontief's inverse matrix to compute the following elements:

I. 　　$\sum_j b_{ij}$ = Sum of the elements of the $j^{th}$ column of the Leontief inverse matrix **B**;
II. 　　$\sum_i b_{ij}$ = Sum of the elements of the $i^{th}$ row of the Leontief inverse matrix **B**;
III. 　$\sum_i \sum_j b_{ij}$ = Total sum of Leontief inverse matrix **B**'s elements;
IV. 　$B^*$—Average value of all elements of Leontief inverse matrix **B**:

$$B^* = \frac{\sum_i \sum_j b_{ij}}{n^2} \tag{10}$$

where: $n$ = Number of sectors. Thus, the forward and backward linkage indices are defined as follows:

V. 　　Backward Linkage Indices

$$U_j = \frac{\frac{\sum_j b_{ij}}{n}}{B^*} \tag{11}$$

VI. 　Forward Linkage Indices

$$U_i = \frac{\frac{\sum_i b_{ij}}{n}}{B^*} \tag{12}$$

where: $\frac{\sum_j b_{ij}}{n}$ = Mean value of the $j^{th}$ column of the Leontief inverse matrix **B**; $\frac{\sum_i b_{ij}}{n}$ = Mean value of the $i^{th}$ row of the Leontief inverse matrix **B**. $n$ = Number of sectors.

The Backward Linkage Indices ($U_j$), or dispersion power index of sector $j$, refers to the ratio of the mean of the elements of Column $j$ of Leontief's inverse matrix and the average of all elements of this matrix. The Forward Linkage Indices ($U_i$), also known in the literature as the dispersion sensitivity index of sector $i$, is the ratio of the mean of the elements of Row $i$ of Leontief's inverse matrix and the average of all elements of this matrix.

$U_i$ describes the direct and indirect impacts on sector $i$ resulting from the total variation (direct, indirect, and induced) caused by a water unit ($m^3$) in the final water demand of the other productive sectors. $U_j$ is the total variation (direct, indirect, and induced) in the production of the entire economic structure necessary to meet a unit variation in the final water demand of sector $j$.

The direct effect is the effect caused by the sector itself; the indirect effect is the effect generated by the sector on other sectors due to their interactions; and the induced effect is the effect generated in the sector due to household consumption on the regional economy's final water demand, given the level of household income obtained by the workforce.

The forward or backward linkage indices with values greater than the unit indicate sectors with a linkage level above the regional average. Therefore, these sectors are key sectors that are considered more dynamic within the regional economic structure. These sectors have strong linkage effects in terms of the flow of goods and services; that is, they are key sectors for the growth of the economy.

If $U_j > 1$, a strong backward link of sector $j$ exists, as it indicates that a unit change in the final water demand for sector $j$ creates an above-average increase in the economy

overall. $U_i > 1$, in turn, symbolises a strong forward link, indicating that unitary changes in the final water demands of all sectors create an above-average increase in sector *i*.

### 2.2.4. Elasticity of Water Consumption

To find the main sectors regarding water consumption, an array of intersectoral elasticities was constructed on the final water demand of Ceará's productive sectors, following the methodology described by [61].

The elasticity of water consumption points to the sector with greater power to propagate the impact from demand shocks on the regional economic structure due to a 1% variation in final water demand.

Consider Γ a scalar that denotes the total water use by the productive system and τ′ a row vector of the water use per unit of sector product. From Leontief's model, one can write:

$$G = \tau' X^* = \tau' (I - A^*)^{-1} Y^* \tag{13}$$

If water use depends on the final water demand of the economy, one can write:

$$\Delta G = \tau' \Delta X^* = \tau' (I - A^*)^{-1} Y^* \gamma \tag{14}$$

where $Y^*$ corresponds to the final water demand in the economy, and γ is a scalar that represents the proportional increase in final water demand. Let S be a vector of the participation of sectoral final water demands in their respective effective productions:

$$S = \left(\hat{X}^*\right)^{-1} Y^*, \text{ or } Y^* = S\hat{X}^* \tag{15}$$

By replacing Equation (15) in Equation (14), one obtains:

$$\Delta G = \tau' (I - A^*)^{-1} S\hat{X}^* \gamma \tag{16}$$

Dividing Equation (16) by Γ:

$$G^{-1} \Delta G = G^{-1} \tau' (I - A^*)^{-1} S\hat{X}^* \gamma \tag{17}$$

where $G^{-1} \Delta G$ shows the total increase in water consumption to an increase in final water demand, that is, the elasticity of Γ to the final water demand. However, this expression does not bring any additional information, given the linear nature of the model, because $G^{-1} \Delta G = \gamma$. Therefore, it is necessary to breakdown the elasticity.

The first step in the elasticity disaggregation is to transform Equation (17). Consider a vector of the final distribution of water among the n productive sectors of the economy, such that $\sum_{i=1}^{n} d_i = 1$. Thus, the vector of the sectoral consumption coefficients τ′ can be written as:

$$\tau' = Gd' \left(\hat{X}^*\right)^{-1} \tag{18}$$

Replacing Equation (18) in Equation (17):

$$G^{-1} \Delta G = d' \left(\hat{X}^*\right)^{-1} (I - A^*)^{-1} S\hat{X}^* \gamma \tag{19}$$

Considering that:

$$(I - D)^{-1} = \left(\hat{X}^*\right)^{-1} (I - A^*)^{-1} \hat{X}^* \tag{20}$$

According to [21], when any two matrices, P and Q, are connected by the ratio $P = MQM^{-1}$, they are said to be similar and are expressed by $P \approx Q$. Then, the product on the right hand side of Equation (16) becomes: $(I - D)^{-1} \approx (I - A^*)^{-1}$; that is, $(I - D)^{-1}$ can be understood as the approximate value of the total (direct and indirect) needs for the

production of goods and services in the economy, which are usually obtained from the matrix $(I - A^*)^{-1}$.

With the diagonalisation of vector S, one can obtain the proportional variation of water consumption of each economic sector with a proportional change in final water demand. From Equations (19) and (20), we obtain:

$$\varepsilon' = d'(I - D)^{-1}\hat{S}\gamma \tag{21}$$

where $\varepsilon'$ represents the proportional variation of the sector's water consumption with a proportional change in final water demand, $\hat{S}$ is the vector S diagonalized and d' is a vector that informs the apportionment of the final water demand among the n productive sectors of the economy.

Omitting $\gamma$ and diagonalising the vector d', one has:

$$G^y = \hat{d}(I - D)^{-1}\hat{S} \tag{22}$$

where $\tau_{ij}^y$ is the characteristic element of the $G^y$ matrix and expresses the percentage increase in the final water consumption of sector $i$ in response to a 1% change in the final water demand of sector $j$ and, $\hat{d}$ ís the vector d' diagonalized. It can be interpreted as elasticity, given that the sum of the elements of the sector Column j expresses the percentage variation in water consumption in the economy with a 1% change in final demand of sector $j$.

As $\tau_{ij}^y$ is an element of the $G^y$ matrix, one can define:

$$\Gamma_{D_{*j}} = \sum_{i=1}^{n} \tau_{ij}^y/n \; (i = 1, 2, \ldots, n) \tag{23}$$

$$\Gamma_{D_{i*}} = \sum_{j=1}^{n} \tau_{ij}^y/n \; (j = 1, 2, \ldots, n) \tag{24}$$

$$\Gamma_{T_{ij}} = \sum \tau_{ij}^y/n \; (j = 1, 2, \ldots, n) \tag{25}$$

Thus, the calculation of elasticities of water demand provides information in a matrix. Each element of a given column shows the contribution of the direct and indirect impact resulting from the 1% increase in the final water demand of production in each other sector of the economy. The sum of the elements of a given column shows the total impact on water demand of all sectors by purchasing resources from other sectors, generated by a one percentage point increase in the final water demand of a given sector. Similarly, the sum of each row represents the distributive impact of water demand generated by the sale of resources in one sector, when the final water demand of the other sectors increases by a percentage point.

According to [61], the impact generated by the 1% variation in the final water demand is divided into two components: the distributive and the total impact. The distributive impact is expressed by the value of the average of the rows. It presents the increase in water consumption in sector '$i$' due to a 1% increase in forward linkages in the final water demand experienced by all sectors of the economy. The average value of the columns expresses a 1% increase in the backward linkage in the final water demand in sector '$j$'. Both these are given by Equations (23) and (24), respectively.

The total impact is obtained by the global average of rows and columns. It shows the percentage increase in energy consumption caused by a 1% increase in final water demand in the economy. It is given by Equation (25). Defining $\Gamma_D$, and $\Gamma_T$ as the median values of total and distributive impacts respectively, [61] adopt the classification established in Table 2.

The sectors in Quadrant I have their water consumption determined, in part, by the demand of the other sectors because the distributive impact is greater than the median of the economy.

The sectors in Quadrant II are the key sectors because they have a greater total and distributive effect than the median values of the economy. That is, they are induced to consume water by increasing demand from other sectors; at the same time, by increasing their own demand, they induce other sectors to consume water.

**Table 2.** Classification of sectors by the elasticity of water demand.

|  | $\Gamma_{D_i} < \Gamma_T$ | $\Gamma_{D_i} > \Gamma_T$ |
|---|---|---|
| $\Gamma_{D_j} > \Gamma_T$ | Demand-relevant sectors from other sectors' point of view I | Key sectors II |
| $\Gamma_{D_j} < \Gamma_T$ | Non-relevant sectors III | Relevant sectors from the point of view of your demand IV |

Source: [61].

Quadrant III shows the least relevant sectors concerning water consumption. Finally, Quadrant IV shows the sectors with high influence on water demand.

## 3. Results

Water is an essential resource for the sustainable development of all regions. However, in Ceará, this is a scarce resource due to its geographical characteristics; it is located in the semiarid region of Brazil where the low water availability is a result of the combination of several factors, like low precipitation rates (less than 900 mm), high evaporation rates (greater than 2000 mm), irregularity of the precipitation regime (frequent and some-times multiannual droughts), and an unfavourable hydrogeological context (80% of the territory is on crystalline rock, with a shallow soil layer and few underground water resources) [62].

Analysing the water flows in the regional economy can help the planning and management of water resources. Water is the main production input. Its level of demand for economic activities can affect regional water security, thereby affecting the production of goods and services, and in turn the level of economic activities.

Here, we present the results of the water flows by applying our methodology of simple multipliers of production, employment, and income to evaluate the efficiency of water use, and the analysis of the Rasmussen and Hirschman indices and the calculation of the elasticity of water demand to evaluate the 'Forward and Backward' linkages of water resource consumption. Our aim is to observe the sectors with a higher degree of influence and impact on direct and intermediate water consumption in Ceará's economy.

### 3.1. Direct Water Consumption through Production, Employment, and Income Multipliers

Direct water consumption and the multipliers of production, employment, and income in terms of water resource flow are presented in Table A1 of the Appendix A.

According to RIOM-WR, direct water consumption by Ceará's economic sectors adds up to a water demand of 1.33 trillion m$^3$. Among these sectors, agriculture has the highest direct consumption at 62.08%, followed by the service sector at 30.50%, and finally industry at only 7.42%.

The agricultural sector deals in products related to permanent and temporary crops, livestock, fishing, and plant extraction. These activities, in general, consume essential raw water in their production processes. However, despite being the sector with the highest level of water consumption in Ceará, it has the lowest level of efficiency in water use in the productive processes among the sectors analysed.

The multipliers of production, employment, and income are, respectively, R\$ 17.09/m$^3$, R\$ 3.01/m$^3$, and 0.80 Job/1000 m$^3$ when compared to the other sectors of Ceará's economy. The manufacturing of motor vehicles, trailers, and bodywork, and other industries in the transportation equipment sector has the highest multipliers of production, income,

and employment per m$^3$ of water (R\$ 20,986.41/m$^3$ water; R\$ 4893.68/m$^3$ water; 315.80 Job/1000 m$^3$ water) (Appendix A, Table A1).

The multipliers of the agricultural sector represent only 0.08%, 0.06%, and 0.25% of that of the manufacturing of motor vehicles, trailers, and body and other transport equipment sector, respectively.

Thus, the latter sector deserves greater attention from water resource managers to reduce water consumption and achieve higher levels of productivity, given the difficulties this sector faces regarding water scarcity for the development of its activities.

Other sectors with high production multipliers were manufacture of machinery and equipment (R\$ 12,435.03/m$^3$ water); and manufacture of computer equipment, electronic and optical products, machinery, and appliances and electrical materials (R\$ 11,723.20/m$^3$ water), which are presented in Table A1 of the Appendix A.

Manufacture of motor vehicles, trailers, and bodywork and other transport equipment presented a production multiplier of approximately R\$ 20,986.41, which indicates that an increase in water volume of 1 m$^3$ in the final demand of this sector would lead to an increase in the production of the sector of R\$ 20,986.41.

In this case, the sector presented an income multiplier of around R\$ 4893.68. This indicates that the increase of a 1 m$^3$ water unit in the final demand of this sector would lead to an increase in the sector's production of R\$ 4893.68 in the wage mass of workers employed in the economy.

High income multipliers were observed for these economic sectors: manufacture of machinery and equipment (R\$ 3191.50/m$^3$ water), and manufacture of computer equipment, electronic and optical products, machinery, appliances, and electrical materials (R\$ 2091.06/m$^3$ water) (Appendix A: Table A1).

Other sectors with high employment multipliers were manufacture of machinery and equipment (277.62 Jobs/1000 m$^3$ water), and manufacture of pharmaceutical and pharmaceutical products (150.67 Jobs/1000 m$^3$ water) (Appendix A: Table A1).

This result indicates that the 1000 m$^3$ increase in final demand in the manufacturing of motor vehicles, trailers, and bodywork, and other transport equipment sectors would lead to an increase of 315 units of jobs generated by this sector and its threads with other sectors.

The manufacture of motor vehicles, trailers, and bodywork, and other transport equipment, manufacture of machinery and equipment, manufacture of computer equipment, electronic and optical products, machinery, appliances, and electrical materials, and manufacture of pharmaceutical and pharmaceutical products have the highest multipliers of production, employment, and income; however, together, they only account for 0.01% of the volume of water consumed by Ceará's economic sectors (Appendix A: Table A1).

These sectors are part of Ceará's industrial park and reflect the dynamics that the sector presents in its production processes associated with low levels of water consumption and higher added value in the goods and services offered to the regional economy.

In 2013, Ceará's industrial sector consumed 98,816,204 m$^3$ of water in its production process. The main economic sectors in terms of direct water consumption in production processes are manufacture of textile products, clothing items and accessories, footwear and leather goods (43,088,462 m$^3$), extractive industries (18,317,716 m$^3$), beverage manufacturing (11,406,933 m$^3$), and food manufacturing (7,914,521 m$^3$). These together account for 6.06% of the total water consumed directly by all economic sectors.

Among the service sectors, the administration, defence, education, public health, social security, private education and health, arts, culture, sport and recreation, and other service activities are important. Although these sectors have lower production, employment, and income multipliers than industrial sectors, they have a high level of direct water consumption of 206,024,667 m$^3$ (15.47%).

Notably, these service delivery sectors are generally located in urban areas and are supplied by better quality treated water distribution networks. This is different from the agricultural sector that uses raw water in its production processes.

*3.2. Intermediate Water Consumption and Its Linkages through Rasmussen and Hirschman Indices*

In terms of water governance, the analysis of intermediate water consumption by economic activities represents a management and planning mechanism oriented to water security due to regional economic growth [63].

According to the RIOM-WR, the intermediate water consumption by Ceará's economic sectors amounts to 193.47 billion m$^3$. The agricultural sector consumes 37,479,301 m$^3$ (19.37%), the industrial sector consumes 25,047,820 m$^3$ (12.94%), and the service sector consumes 67.68% (130,944,545 m$^3$) (Appendix A: Table A2).

Among these sectors, administration, defence, education, and public health and social security with 60,477,024 m$^3$ (31.26%), agricultural with 37,479,301 m$^3$ (19.37%), and private education and health with 14,905,943 m$^3$ (7.70%) have the highest intermediate water consumption (Appendix A: Table A2).

This analysis allows us to see and measure a hidden component of total water use within production processes. We can better identify intersectoral water flows, especially the scale and manner in which intermediate water consumption occurs between and within economic activities [22,63].

In this sense, intermediate consumption represented a significant part of the total water consumption in the state. This demonstrates that Ceará, despite being in a semiarid region with water scarcity, has an economic structure based on sectors that consume large amounts of water. Therefore, analyses that only consider the direct water use in each sector generate erroneous conclusions, considering that these sectors indirectly consume a sufficiently important amount of water to threaten the sustainability of the state's water supply.

This type of consumption can be better qualified by the Rasmussen and Hirschman indices on the 'backward and forward linkages' connections. These indices establish the sectors that have the greatest power to propagate indirect water consumption within the economy through demand and intersectoral supply of goods and services in the form of productive resources, considering direct, indirect, and induced effects.

When we jointly analysed the forward and backward linkages indices applied to the economic structure of water consumption among Ceará's productive sectors, we found that only the trade and repair of motor vehicles and motorcycles sector has a dynamic water consumption above the state average (Figure 4, Appendix A: Table A2).

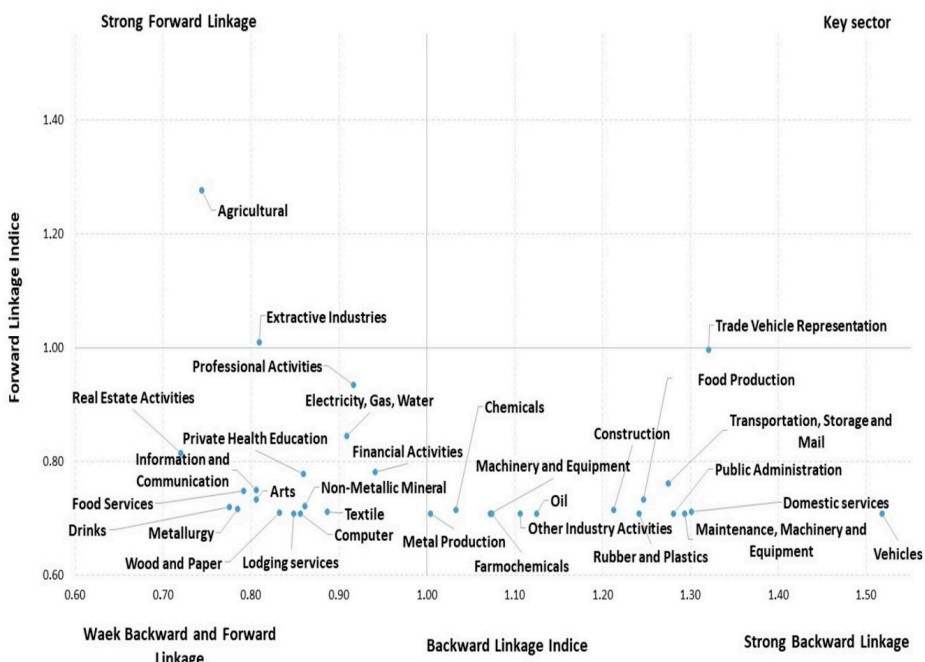

**Figure 4.** Rasmussen and Hirschman linkages indices (direct, indirect, and induced effects) of water flows in forward and backward linkages in Ceará.

This occurs because this sector is the second most important sector within Ceará's macro service sectors with 18.95% of the Gross Added Value. This sector has 9,703,543 m$^3$ of intermediate water consumption in the form of resources in its production processes. This represents 5.01% of the total volume of intermediate water consumption among all of Ceará's sectors.

Notably, the trade and repair of motor vehicles and motorcycles has stronger forward linkages with the following sectors, through the provision of these services by their Commercial Representatives and Agents of Commerce: Professional, Scientific, and Technical Activities, Administrative and Complimentary Services, Real Estate Activities, Agriculture, Financial Activities, Insurance, and Related Services. Furthermore, the sector is strongly impacted by household consumption (Appendix A: Table A2).

We found that the agricultural sector, despite being recognised as a sector that consumes a high amount of water in its production processes, is not identified as a key sector in terms of water flows. However, it proved to be an important sector in terms of forward linkages in water consumption. Furthermore, it had strong connectivity with these sectors: trade and repair of motor vehicles and motorcycles, and the manufacture of food products. The agriculture sector is also impacted by household consumption, as it induces the production of goods and services in this sector.

(a)     Rasmussen and Hirschman Backward Indices

The sectors with strong backward linkages in terms of water flows were manufacture of motor vehicles, trailers, and bodywork and other transport equipment (1.5180), trade and repair of motor vehicles and motorcycles (1.3206), and domestic services (1.3009). These sectors have intermediate water consumption at 149,601 m$^3$ (0.08%), 9,703,543 m$^3$ (5.02%), and 7,968,170 m$^3$ (4.12%), respectively. Only the manufacture of motor vehicles, trailers and bodywork and other transport equipment sector has a lower representation in total intermediate water consumption (Appendix A: Table A2).

Beijing's international trade in China is examined by [64], who point out that the following productive sectors have the highest backward linkage index and are more related to the service sector: food processing and tobacco; construction; synthetic technique services; hospitality and gastronomy; information transmission, software and information technology services. This indicates that to support their production processes, these sectors need large imports with water resources incorporated into their goods and services.

(b) Rasmussen and Hirschman Forward Indices

The forward linkage index in terms of water flows indicates two sectors with strong linkages: agriculture (1.2780) and extractive industry (1.0089). These data indicate that these sectors offer the most virtual water through their goods and services to other sectors (Appendix A: Table A2). These two sectors interact with other productive sectors with a level of influence above the state average.

According to [13,65], the agricultural sector has a high power to spread impacts on the economy caused by shocks to agroindustry and the repercussions from the agricultural sector as supplier of resources. This is through interactions between the agricultural sector and animal slaughter agro-industries, processing of meat products, production of dairy products, and fishing.

The agricultural sector is highly important for the Brazilian economy according to [66]. This is because the sector is a buyer of goods and services, and a generator of resources for other sectors. This highlights its role as a supplier of raw materials for the development of non-agricultural sectors, and its importance for the consumer market and industrialised products.

Our analysis reveals that the agriculture sector, production and supply of electricity, steam and hot water, metallurgy, foundry and pressing, chemicals, and trade were the sectors that have the highest forward linkage indices in terms of water demand through the interactions that these productive sectors have in Ceará's local economy. Therefore, this

produces a multiplier effect of the impact of their processed products on the other sectors within the regional economic structure.

Regarding forward linkages, household consumption has a great induced effect (8.3247) on sectoral economic production. This is because households allocate a large part of their income for the acquisition of goods and services in the economy. They obtain income by making their labour available to companies.

These results for the forward linkage index suggest that sectors that have a high level of forward linkages consume more physical water than virtual water in their production processes. Therefore, an increase in production within these sectors tends to exert greater pressure on existing water sources in the region, such as dams and wells, to meet this increase in demand.

Thus, effective monitoring of both physical and virtual water flows enables better management of regional water availability, by relating the forces of demand for water with the existing water supply in a way that is compatible with the multiple interests of its users and their managers.

Therefore, the knowledge of water flows generated by economic activities improves the governability of water supply, by enabling the analysis of the support capacity of the physical structure of storage and distribution of water in the face of the demand forces exerted by economic activities.

(c) Impact Multiplier Effect: Direct, Indirect, and Induced

We also observed the direct, indirect, and induced average effects generated by water flows through the Rasmussen and Hirschmann indices for the forward linkages of all economic sectors of Ceará. We found that these are more influenced by the indirect effects generated by the sectoral interrelations existing in the regional economy, and contribute on average 90.80% of the total effect generated in the regional economic structure.

This is because Ceará's economic sectors are induced to consume more water due to their interactions with other sectors when the former offer goods and services in the economy. Intersectoral demand puts pressure on water consumption in their production systems.

The trade and repair of motor vehicles and motorcycles sector is a key sector in terms of water flows. We observed that this sector's total effect had the following composition in terms of direct, indirect, and induced effects: 22.90%, 71.38%, and 5.72%, respectively. Clearly, the indirect effects for the forward linkages are stronger.

The mean direct, indirect, and induced effects generated by the water flows were 4.34%, 71.19%, and 24.48%, respectively. Thus, indirect effects also have greater participation in the total effect, but not at the same intensity as in forward linkages.

This may be because of greater participation of the induced effects generated by household consumption in the total effect on the regional economy. In turn, this may be because part of the production generated by economic sectors is intended to meet household needs in their backward linkages.

The economic sectors with the largest multiplier effect via backward linkages were manufacture of motor vehicles, trailers, and bodywork, and other transport equipment (2.1459), trade and repair of motor vehicles and motorcycles (1.8669), domestic Services (1.8390), and maintenance, repair and installation of machinery and equipment (1.8285) (Table A3 of the Appendix A).

Sectors with the greatest multiplier effect via forward linkages were agriculture (1.8066), extractive industries (1.4262), and trade and repair of motor vehicles and motorcycles (1.4088). The induced effect generated by household consumption (11.7681) had a great weight in the total effect on the production of all sectors of Ceará's economy (Appendix A: Table A4).

Notably, the induced effect caused by household consumption mainly stimulates the forward linkage effects on Ceará's economic structure. Furthermore, it affects the agricultural sector that is well connected with the industrial and services sectors. Thus,

household consumption has a high multiplier effect that directly and indirectly stimulates the structure of the economy and the demand from its sectors.

*3.3. Water Flows: Elasticity*

According to [61], the calculation of demand elasticities through the Regional IOM provides information on the sensitivity of each sector in the face of a demand shock from the other sectors.

Each element of a given column of the matrix shows the distributive impact through the backward linkages that each sector has due to a 1% increase in the final water demand from the production of the other sectors; that is, the sum of the elements of a given column shows the distributive impact through the backward linkage, in terms of virtual water demand, within the regional economic structure in response to a percentage point increase in the final water demand of the other sectors.

Analogously, the sum of each row represents the forward distributive impact generated in one sector when the final water demand of the other sectors increased by a percentage point.

Thus, the sectors with the highest demand elasticity in terms of backward linkages (Appendix A: Table A5) were: manufacture of motor vehicles, trailers, bodywork, and other transport equipment ($\Gamma_{D_j}$ = 1.0105), trade and repair of motor vehicles and motorcycles ($\Gamma_{D_j}$ = 1.0077), and domestic services ($\Gamma_{D_j}$ = 1.0075).

These sectors have a higher level of sensitivity to water consumption in the coefficients of direct consumption of goods and services within their production process; that is, when the demand of the other sectors increases, these are the sectors that are pressured to demand more water resources in a productive backward linkage within the regional economic structure.

In an analysis of the elasticity of demand for forward linkages, agriculture ($\varepsilon_i$ = 1.0081) and extractive industries ($\varepsilon_i$ = 1.0043) were the economic sectors with a higher increase in the level of water consumption due to the increased water demand in the other sectors. Thus, the consumption of families also showed a strong sensitivity regarding the elasticity of the demand for water within the regional economic structure ($\varepsilon_i$ = 1.1078).

In this case, water demand in these sectors is more influenced by their own demand than by the demand of other sectors. This is because these are sectors that supply inputs to the other sectors, as they have a higher level of forward linkage in their production process.

The trade and repair of motor vehicles and motorcycles ($\Gamma_{D_j}$ = 1.0077; $\Gamma_{D_i}$ = 1.0032), is Ceará's key sector regarding the elasticity of water demand together with household consumption ($\Gamma_{D_j}$ = 0.9997; $\Gamma_{D_i}$ = 1.0067) (Table 3). This is because both increase pressures on water consumption and are pressured to consume due to the increased water demand from the other sectors, due in turn to their forward and backward linkages within Ceará's economic structure.

The other sectors of the economy are considered not to be relevant because have they have forward and backward distribution impacts below the overall state average for a 1% increase in final water demand.

**Table 3.** Classification of sectors by the elasticity of water demand.

|  | $\Gamma_{D_i} < \Gamma_T$ | $\Gamma_{D_i} > \Gamma_T$ |
|---|---|---|
| $\Gamma_{D_j} > \Gamma_T$ | I<br>Manufacture of vehicles; Domestic services; Maintenance, repair, and installation of machinery and equipment; Public administration; Transport, storage, and mail; Prod. Food; Rubber and plastic products; Construction; Refining of petroleum and coke and alcohol and other biofuels; Other industrial activities; Pharmaceutical and pharmaceutical products; Machinery and Equipment | II<br>Trade and repair of motor vehicles and motorcycles, Household consumption |
| $\Gamma_{D_j} < \Gamma_T$ | III<br>Other sectors | IV<br>Agriculture, and Extractive industries |

## 4. Conclusions

This study sought to reveal the most relevant sectors in the economic structure, from the perspective of direct and indirect use of water. We also sought to perform sectoral and intersectoral analyses of the economic impact of consumption and exchange of water resources. The aim was to better qualify the typology of water use and its economic efficiency to support the formulation of strategies by decision-makers.

According to the RIOM-WR, direct water consumption by Ceará's economic sectors adds up to a total water demand of 1.33 trillion m$^3$. Among these sectors, agriculture has the highest direct consumption of raw water with 62.08%, the service sector accounts for 30.50%, and finally the industry sector consumes only 7.42%.

We found that the agricultural sector, despite being recognised as a sector that consumes a high amount of water resources in its production processes, was not a key sector in terms of water flows. Nevertheless, it had one of the lowest levels of water use efficiency among the sectors analysed. It was an important sector in terms of forward linkages in water consumption. Furthermore, it had strong connectivity with these sectors: households, trade and re-pair of motor vehicles and motorcycles, and the manufacture of food products. Lastly, it was not possible to perform a disaggregated analysis of the agricultural sector considering that our study was based on Ceará's Input-Output Matrix. The matrix provides only aggregate data for the agricultural sector. Therefore, we could not undertake more detailed analysis of the sector's individual productive activities.

According to the RIOM-WR, intermediate water consumption by the economic sectors is 193.47 billion m$^3$. The agricultural sector consumes 37,479,301 m$^3$ (19.37%), the industry sector consumes 25,047,820 m$^3$ (12.94%), and the service sector consumes 130,944,545 m$^3$ (67.68%).

Intermediate consumption is strongly influenced by the interrelations of the economic sectors. According to the analysis of the Rasmussen and Hirschman indices on forward and backward linkages together, only the following key sectors were identified in terms of water flows: trade, and repair of motor vehicles and motorcycles. This is the sector has strong linkage effects, in terms of water flow, through the production and commercialisation of goods and services within the regional economy. Furthermore, it has a capacity, above the state average, to contribute to the growth of water consumption in the regional economy.

The results for the forward linkage index suggest that sectors with a high level of forward linkages consume more physical water than virtual water in their production processes. Therefore, an increase in production of these sectors tends to exert greater pressure on existing water sources in the region, such as dams and wells, to meet this increase in demand.

The sectors with strong backward linkages in terms of water flows were the manufacture of motor vehicles, trailers, and bodywork, and other transport equipment (1.5180), trade and repair of motor vehicles and motorcycles (1.3206), and domestic services (1.3009). These sectors have the following intermediate water consumption at 149,601 m$^3$ (0.08%), 9,703,543 m$^3$ (5.02%), and 7,968,170 m$^3$ (4.12%), respectively. They have a strong dependence on resources generated by productive activities belonging to the sector itself or other sectors of the economy.

During the analysis of the impact multipliers in the forward and backward linkages, the trade and repair sector of motor vehicles and motorcycles presented the highest multipliers (1.41 forward and 1.86 backward). This indicates that the direct consumption of a 1 m$^3$ water unit per this sector would generate a final water demand of 1.41 m$^3$ and 1.86 m$^3$ in direct and rear links, respectively.

In the analysis of water use efficiency among economic sectors, the following sectors had the highest multipliers of production, employment, and income: manufacture of motor vehicles, trailers, bodywork, and other transport equipment; manufacture of machinery and equipment; manufacture of computer equipment, electronic and optical products, machinery, appliances, and electrical materials; and manufacture of pharmaceutical and pharmaceutical products. However, together these sectors only account for 0.01% of the total water consumed by Ceará's economic sectors.

The elasticity of water demand showed that the trade and repair of motor vehicles and motorcycles sector ($\Gamma_{D_j}$ = 1.0077; $\Gamma_{D_i}$ = 1.0032), and household consumption ($\Gamma_{D_j}$ = 0.9997; $\Gamma_{D_i}$ = 1.0067) are the key sectors within Ceará's economy. That is, these sectors are more sensitive to water demand because both put pressure on water consumption, and are pressured to consume water. This is because of the increased water demand from other sectors due to their forward and backward linkages in Ceará's economic structure.

Among the economic sectors analyzed, the trade and repair of motor vehicles and motorcycles sector can be considered as a key sector by presenting Rasmussen and Hirschman linkage indices, production, employment, and income multipliers, and elasticity of water demand above the state average of Ceará. The sector has a direct water consumption level of 12,019,037 m$^3$ (0.90%) and an intermediate water consumption level of 9,703,543 m$^3$ (5.02%). Notably, this sector has stronger forward linkages with the following through the provision of services by their commercial representatives and agents of commerce: Professional, Scientific, and Technical Activities; Administrative and Complimentary Services; Real Estate Activities; Agriculture; and Financial Activities, Insurance, and Related Services. The sector is also strongly impacted by household consumption.

The consumption of families also has a powerful influence over the demand for water in the productive sectors of Ceará's regional economy. This is because the level of household consumption can generate great effects of both forward and backward linkages with the other sectors. This makes household consumption a dynamic factor within the regional economic structure.

The key sectors of the economy, in terms of water flows, indicates to decision-makers which sectors have the highest capacity for linkage within the economy, i.e., it reveals the ability of these sectors to propagate the effects of investments or external shocks on the economic structure in a proportion above the average of all productive sectors, besides identifying the economic sectors that directly or indirectly consume a greater volume of water within Ceará's economic structure.

For example, one can investigate which economic sectors should be evaluated during a water crisis based on the integration of water supply with economic planning. Thus, both the balance of water flow and efficient water use in the production processes of goods and services are important for regional development, given their impacts on income, employment, and water security.

Furthermore, effective monitoring of both physical and virtual water flows enables better management of available regional water by relating the forces of water demand and existing supply in a way that it is compatible with the multiple interests of its users and their managers.

Thus, the evolution of regional economic growth must be in line with the existing water supply infrastructure. The knowledge of both physical and virtual water flows, associated with direct water consumption by various economic sectors, helps in the analysis of the hydraulic support capacity of water distribution networks (reservoirs, canals, pipelines, and treatment stations), and the management of the volume of water needed to meet the demands of Ceará's economic structure.

An efficient water supply system should be associated with the monitoring of water demand information in line with its supply. It should allow us to evaluate complex situations of calibration of pressures, flows, water levels of reservoirs, manometric heights, pumping flows, etc. The overall aim should be an operational model that seeks to ensure economic and safe water supply in urban and rural areas, without compromising the supply and quality of water resources.

Our methodology generated useful information to support the management and formulation of policies regarding the allocation of water resources. While the estimated impacts are specific estimates in time and space, they are not associated with measures of uncertainty in their estimates. This is a limitation of our study.

Future studies should discuss policy measures to improve the management of water demand associated with the development of economic activities. Furthermore, it is im-

portant to consider a greater opening of the matrix to the agricultural sector to improve the analysis of its supply chain, considering that this is one of the sectors that directly consumes the most water resources directly.

We also suggest breaking down the indices of the impact of forward and backward linkages and the analysis of the overflow effect of virtual water demand by interregional trade to study the sensitivity of aggregate demand in the economy to external shocks of supply and demand.

**Author Contributions:** Conceptualization, methodology, and validation, R.B.S., S.M.O.d.S. and W.d.L.P.; formal analysis, investigation, writing—original draft preparation, R.B.S. and S.M.O.d.S.; selection of analysis techniques, R.B.S. and W.d.L.P.; writing—review and editing: F.d.A.d.S.F. and S.M.O.d.S.; supervision: F.d.A.d.S.F. and S.M.O.d.S. All authors have read and agreed to the published version of the manuscript.

**Funding:** The research was partially supported by grants from the Conselho Nacional de Desenvolvimento Científico e Tecnológico—Brasil (CNPq), Fundação Cearense de Apoio ao Desenvolvimento Científico e Tecnológico (Project, Nº 11098079/2019) and Coordenação de Aperfeiçoamento de Pessoal de Nível Superior—Brasil (CAPES)—financing code 001.

**Institutional Review Board Statement:** Not applicable.

**Informed Consent Statement:** Not applicable.

**Data Availability Statement:** The data presented in this study is available on request from the corresponding author.

**Acknowledgments:** The authors thank the Conselho Nacional de Desenvolvimento Científico e Tecnológico—Brasil (CNPq), Brazil and Fundação Cearense de Apoio ao Desenvolvimento Científico e Tecnológico for the financial support received for this research. We also thank the Institute of Research and Economic Strategy of Ceará (IPECE) for the technical support and availability of data.

**Conflicts of Interest:** The authors declare no conflict of interest.

## Appendix A

**Table A1.** Direct water consumption, and sector multipliers of production, income, and employment in terms of water flows.

| Economic Sectors | Direct Water Consumption (DWC) (m$^3$) | DWC (%) | Production (R\$/m$^3$ Water) | Income (R\$/m$^3$ Water) | Job (Job/1000 m$^3$ Water) |
|---|---|---|---|---|---|
| | | | **Multipliers** | | |
| 0101 Agricultural | 82,674,089,052 | 62.08 | 17.09 | 3.01 | 0.80 |
| 0201 Extractive industries | 7,921,305,183 | 5.95 | 205.85 | 50.15 | 3.25 |
| 0301 Manufacture of food products | 7,448,717,530 | 5.59 | 1023.47 | 144.45 | 9.87 |
| 0302 Beverage manufacturing | 5,232,444,991 | 3.93 | 246.87 | 38.51 | 2.58 |
| 0303 Manufacture of textile products, clothing and accessories, footwear, and leather goods | 4,680,501,360 | 3.51 | 260.69 | 75.19 | 7.53 |
| 0304 Manufacture of wood products, excluding furniture, pulp, paper, and paper products and printing and reproduction services of recordings | 4,308,846,262 | 3.24 | 493.46 | 93.77 | 7.65 |
| 0305 Refining of petroleum and coke and alcohol and other biofuels | 4,151,000,299 | 3.12 | 1449.10 | 92.17 | 3.05 |

**Table A1.** *Cont.*

| Economic Sectors | Direct Water Consumption (DWC) (m³) | DWC (%) | Multipliers | | |
|---|---|---|---|---|---|
| | | | Production (R\$/m³ Water) | Income (R\$/m³ Water) | Job (Job/1000 m³ Water) |
| 0306 Manufacture of chemicals | 3,725,859,123 | 2.80 | 3782.01 | 459.35 | 22.20 |
| 0307 Manufacture of pharmaceutical and pharmaceutical products | 2,526,447,085 | 1.90 | 9482.55 | 2065.37 | 150.67 |
| 0308 Manufacture of rubber, and plastic products | 2,422,950,822 | 1.82 | 6608.94 | 1457.36 | 135.52 |
| 0309 Manufacture of non-metallic mineral products | 1,831,771,618 | 1.38 | 1075.95 | 165.70 | 17.34 |
| 0310 Metallurgy | 1,201,903,717 | 0.90 | 1324.75 | 108.52 | 6.52 |
| 0311 Manufacture of metal products, except machinery and equipment | 1,140,693,301 | 0.86 | 2542.70 | 672.48 | 60.21 |
| 0312 Manufacture of computer equipment, electronic and optical products, machinery, appliances, and electrical materials | 1,006,556,114 | 0.76 | 11,723.20 | 2091.06 | 116.39 |
| 0313 Manufacture of machinery and equipment | 791,452,081 | 0.59 | 12,435.03 | 3191.50 | 277.62 |
| 0314 Manufacture of motor vehicles, trailers, and bodywork, and other transport equipment | 592,450,880 | 0.44 | 20,986.41 | 4893.68 | 315.80 |
| 0315 Other industrial activities | 389,533,665 | 0.29 | 2986.32 | 632.33 | 56.68 |
| 0316 Maintenance, repair, and installation of machinery, and equipment | 218,965,446 | 0.16 | 3322.26 | 547.44 | 40.15 |
| 0401 Electricity and gas, water, sewage, waste management, and decontamination activities | 190,032,220 | 0.14 | 2130.97 | 177.01 | 6.79 |
| 0501 Construction | 165,006,220 | 0.12 | 2563.11 | 539.36 | 54.34 |
| 0601 Trade and repair of motor vehicles, and motorcycles | 155,296,094 | 0.12 | 1770.34 | 495.53 | 61.39 |
| 0701 Transport, storage, and mail | 130,288,371 | 0.10 | 5564.41 | 1501.51 | 113.40 |
| 0801 Lodging services | 113,464,428 | 0.09 | 341.37 | 89.53 | 28.49 |
| 0802 Food services | 42,905,922 | 0.03 | 256.79 | 45.73 | 8.44 |
| 0901 Information and communication | 36,188,131 | 0.03 | 141.30 | 27.62 | 1.16 |
| 1001 Financial, insurance, and related services activities | 29,809,479 | 0.02 | 251.75 | 111.59 | 1.30 |
| 1101 Real estate activities | 13,543,610 | 0.01 | 237.40 | 3.77 | 0.26 |
| 1201 Professional, scientific and technical, and administrative and complimentary services | 12,158,369 | 0.01 | 212.00 | 83.94 | 5.48 |
| 1301 Administration, defence, education, and public health and social security | 7,708,586 | 0.01 | 381.30 | 253.06 | 4.76 |
| 1401 Private education and health | 2,609,629 | 0.00 | 75.04 | 30.73 | 2.17 |
| 1501 Arts, culture, sport, and recreation and other service activities | 2,533,524 | 0.00 | 110.67 | 34.54 | 4.48 |
| 1601 Domestic services | 1,657,177 | 0.00 | 140.51 | 125.71 | 25.27 |

**Table A2.** Intermediate water consumption, and Rasmussen and Hirschman indices for the Backward, and Forward (Direct, Indirect and Induced Effects) of Water Flows, Ceará.

| Economic Sectors | Intermediate Water Consumption (IWC) (m³) | IWC (%) | Backward Linkage Indice | Forward Linkage Indice |
|---|---|---|---|---|
| 0101 Agricultural | 37,479,301 | 19.37 | 0.7439 | 1.2780 |
| 0201 Extractive industries | 415,932 | 0.21 | 0.8097 | 1.0089 |
| 0301 Manufacture of food products | 5,604,857 | 2.90 | 1.2467 | 0.7335 |
| 0302 Beverage manufacturing | 823,196 | 0.43 | 0.7752 | 0.7193 |
| 0303 Manufacture of textile products, clothing and accessories, footwear, and leather goods | 10,081,573 | 5.21 | 0.8865 | 0.7124 |
| 0304 Manufacture of wood products, excluding furniture, pulp, paper, and paper products and printing and reproduction services of recordings | 351,478 | 0.18 | 0.8320 | 0.7101 |
| 0305 Refining of petroleum and coke and alcohol and other biofuels | 583,277 | 0.30 | 1.1251 | 0.7078 |
| 0306 Manufacture of chemicals | 175,482 | 0.09 | 1.0334 | 0.7155 |
| 0307 Manufacture of pharmaceutical and pharmaceutical products | 65,697 | 0.03 | 1.0740 | 0.7080 |
| 0308 Manufacture of rubber and plastic products | 156,222 | 0.08 | 1.2416 | 0.7085 |
| 0309 Manufacture of non-metallic mineral products | 371,243 | 0.19 | 0.8610 | 0.7214 |
| 0310 Metallurgy | 143,490 | 0.07 | 0.7848 | 0.7167 |
| 0311 Manufacture of metal products, except machinery and equipment | 405,007 | 0.21 | 1.0044 | 0.7083 |
| 0312 Manufacture of computer equipment, electronic and optical products, machinery, appliances, and electrical materials | 347,949 | 0.18 | 0.8557 | 0.7078 |
| 0313 Manufacture of machinery and equipment | 118,331 | 0.06 | 1.0723 | 0.7086 |
| 0314 Manufacture of motor vehicles, trailers and bodywork, and other transport equipment | 149,601 | 0.08 | 1.5180 | 0.7081 |
| 0315 Other industrial activities | 353,972 | 0.18 | 1.1063 | 0.7076 |
| 0316 Maintenance, repair, and installation of machinery and equipment | 100,863 | 0.05 | 1.2934 | 0.7078 |
| 0401 Electricity and gas, water, sewage, waste management, and decontamination activities | 903,438 | 0.47 | 0.9091 | 0.8448 |
| 0501 Construction | 3,896,211 | 2.01 | 1.2124 | 0.7150 |
| 0601 Trade and repair of motor vehicles and motorcycles | 9,703,543 | 5.02 | 1.3206 | 0.9966 |
| 0701 Transport, storage, and mail | 3,003,572 | 1.55 | 1.2750 | 0.7616 |
| 0801 Lodging services | 302,228 | 0.16 | 0.8484 | 0.7082 |
| 0802 Food services | 2,387,807 | 1.23 | 0.7912 | 0.7479 |
| 0901 Information and communication | 4,486,112 | 2.32 | 0.8058 | 0.7494 |
| 1001 Financial, insurance, and related services activities | 7,646,770 | 3.95 | 0.9411 | 0.7820 |
| 1101 Real estate activities | 579,917 | 0.30 | 0.7198 | 0.8147 |
| 1201 Professional, scientific and technical, administrative, and complimentary services | 12,919,299 | 6.68 | 0.9163 | 0.9354 |
| 1301 Administration, defence, education, and public health and social security | 60,477,024 | 31.26 | 1.2807 | 0.7082 |
| 1401 Private education and health | 14,905,943 | 7.70 | 0.8595 | 0.7790 |
| 1501 Arts, culture, sport and recreation and other service activities | 6,564,160 | 3.39 | 0.8058 | 0.7331 |
| 1601 Domestic services | 7,968,170 | 4.12 | 1.3009 | 0.7111 |

**Table A3.** Economic sectors with greater impact multiplier effect—Back (Direct, indirect and induced, and total effects) on water flows, Ceará.

| Economic Sectors | Multiplier Effect—Backward | | | |
|---|---|---|---|---|
| | Direct Effect | Indirect Effect | Induced Effect | Total Effect |
| 0314 Manufacture of motor vehicles, trailers and bodywork, and other transport equipment | 0.0987 | 1.0069 | 1.0403 | 2.1459 |
| 0601 Trade and repair of motor vehicles and motorcycles | 0.0721 | 1.0014 | 0.7934 | 1.8669 |
| 1601 Domestic services | | 1.0000 | 0.8390 | 1.8390 |
| 0316 Maintenance, repair, and installation of machinery and equipment | 0.0553 | 1.0036 | 0.7695 | 1.8285 |
| 1301 Administration, defence, education, and public health and social security | 0.0049 | 1.0002 | 0.8054 | 1.8104 |
| 0701 Transport, storage and mail | 0.0870 | 1.0025 | 0.7129 | 1.8024 |
| 0301 Manufacture of food products | 0.6064 | 1.0098 | 0.1460 | 1.7623 |
| 0308 Manufacture of rubber and plastic products | 0.0202 | 1.0013 | 0.7336 | 1.7551 |
| 0501 Construction | 0.0452 | 1.0017 | 0.6670 | 1.7139 |
| 0305 Refining of petroleum and coke and alcohol and other biofuels | 0.4996 | 1.0151 | 0.0758 | 1.5905 |
| 0315 Other industrial activities | 0.0103 | 1.0006 | 0.5529 | 1.5638 |
| 0307 Manufacture of pharmaceutical and pharmaceutical products | 0.0358 | 1.0016 | 0.4809 | 1.5182 |
| 0313 Manufacture of machinery and equipment | 0.0152 | 1.0010 | 0.4997 | 1.5158 |
| 0306 Manufacture of chemicals | 0.0538 | 1.0031 | 0.4039 | 1.4608 |
| 0311 Manufacture of metal products, except machinery and equipment | 0.0066 | 1.0003 | 0.4129 | 1.4198 |
| 1001 Financial, insurance and related services activities | 0.0375 | 1.0013 | 0.2915 | 1.3303 |
| 1201 Professional, scientific and technical, administrative, and complimentary services | 0.0111 | 1.0004 | 0.2837 | 1.2953 |
| 0401 Electricity and gas, water, sewage, waste management and decontamination activities | 0.1667 | 1.0254 | 0.0930 | 1.2851 |
| 0303 Manufacture of textile products, clothing and accessories, footwear and leather goods | 0.0121 | 1.0005 | 0.2405 | 1.2531 |
| 0309 Manufacture of non-metallic mineral products | 0.0353 | 1.0018 | 0.1800 | 1.2171 |
| 1401 Private education and health | 0.0056 | 1.0003 | 0.2092 | 1.2151 |
| 0312 Manufacture of computer equipment, electronic and optical products, machinery, appliances, and electrical materials | 0.0200 | 1.0010 | 0.1886 | 1.2097 |
| 0801 Lodging services | 0.0303 | 1.0012 | 0.1678 | 1.1993 |
| 0304 Manufacture of wood products, excluding furniture, pulp, paper, and paper products and printing and reproduction services of recordings | 0.0148 | 1.0007 | 0.1607 | 1.1762 |
| 0201 Extractive industries | 0.0308 | 1.0024 | 0.1114 | 1.1446 |
| 0901 Information and communication | 0.0705 | 1.0034 | 0.0653 | 1.1391 |
| 1501 Arts, culture, sport and recreation and other service activities | 0.0167 | 1.0010 | 0.1214 | 1.1391 |
| 0802 Food services | 0.0376 | 1.0026 | 0.0783 | 1.1185 |
| 0310 Metallurgy | 0.0405 | 1.0042 | 0.0647 | 1.1095 |
| 0302 Beverage manufacturing | 0.0353 | 1.0022 | 0.0584 | 1.0959 |
| Household consumption | | | 1.0598 | 1.0598 |
| 0101 Agricultural | 0.0069 | 1.0013 | 0.0435 | 1.0516 |
| 1101 Real estate activities | 0.0117 | 1.0004 | 0.0055 | 1.0175 |

**Table A4.** Economic sectors with greater impact multiplier effect—Forward (Direct, indirect and induced, and total effects) on water flows, Ceará.

| Economic Sectors | Multiplier Effect—Forward | | | |
|---|---|---|---|---|
| | Direct Effect | Indirect Effect | Induced Effect | Total Effect |
| Household consumption | | | 11.7681 | 11.7681 |
| 0101 Agricultural | 0.6672 | 1.0082 | 0.1311 | 1.8066 |
| 0201 Extractive industries | 0.4255 | 1.0005 | 0.0002 | 1.4262 |
| 0601 Trade and repair of motor vehicles and motorcycles | 0.3227 | 1.0055 | 0.0806 | 1.4088 |
| 1201 Professional, scientific and technical, administrative, and complimentary services | 0.2855 | 1.0283 | 0.0086 | 1.3223 |
| 0401 Electricity and gas, water, sewage, waste management, and decontamination activities | 0.1554 | 1.0236 | 0.0152 | 1.1942 |
| 1101 Real estate activities | 0.0406 | 1.0082 | 0.1029 | 1.1517 |
| 1001 Financial, insurance and related services activities | 0.0890 | 1.0069 | 0.0096 | 1.1055 |
| 1401 Private education and health | 0.0011 | 1.0000 | 0.1001 | 1.1012 |
| 0701 Transport, storage and mail | 0.0583 | 1.0121 | 0.0062 | 1.0766 |
| 0901 Information and communication | 0.0462 | 1.0032 | 0.0099 | 1.0594 |
| 0802 Food services | 0.0074 | 1.0000 | 0.0499 | 1.0573 |
| 0301 Manufacture of food products | 0.0106 | 1.0012 | 0.0252 | 1.0370 |
| 1501 Arts, culture, sport and recreation and other service activities | 0.0014 | 1.0001 | 0.0349 | 1.0363 |
| 0309 Manufacture of non-metallic mineral products | 0.0195 | 1.0003 | 0.0000 | 1.0198 |
| 0302 Beverage manufacturing | 0.0110 | 1.0001 | 0.0056 | 1.0168 |
| 0310 Metallurgy | 0.0131 | 1.0001 | 0.0000 | 1.0132 |
| 0306 Manufacture of chemicals | 0.0111 | 1.0003 | 0.0000 | 1.0115 |
| 0501 Construction | 0.0105 | 1.0002 | 0.0000 | 1.0107 |
| 0303 Manufacture of textile products, clothing and accessories, footwear, and leather goods | 0.0053 | 1.0000 | 0.0019 | 1.0071 |
| 1601 Domestic services | | 1.0000 | 0.0052 | 1.0052 |
| 0304 Manufacture of wood products, excluding furniture, pulp, paper, and paper products and printing and reproduction services of recordings | 0.0037 | 1.0000 | 0.0000 | 1.0038 |
| 0313 Manufacture of machinery and equipment | 0.0016 | 1.0000 | 0.0000 | 1.0016 |
| 0308 Manufacture of rubber and plastic products | 0.0015 | 1.0000 | 0.0000 | 1.0015 |
| 0311 Manufacture of metal products, except machinery and equipment | 0.0013 | 1.0000 | 0.0000 | 1.0013 |
| 0801 Lodging services | 0.0009 | 1.0000 | 0.0002 | 1.0012 |
| 1301 Administration, defence, education and public health and social security | 0.0010 | 1.0001 | 0.0001 | 1.0012 |
| 0314 Manufacture of motor vehicles, trailers and body work and other transport equipment | 0.0010 | 1.0000 | 0.0000 | 1.0010 |
| 0307 Manufacture of pharmaceutical and pharmaceutical products | 0.0009 | 1.0000 | 0.0000 | 1.0009 |
| 0316 Maintenance, repair and installation of machinery and equipment | 0.0005 | 1.0000 | 0.0001 | 1.0006 |
| 0305 Refining of petroleum and coke and alcohol and other biofuels | 0.0000 | 1.0000 | 0.0005 | 1.0006 |
| 0312 Manufacture of computer equipment, electronic and optical products, machinery, appliances and electrical materials | 0.0005 | 1.0000 | 0.0000 | 1.0005 |
| 0315 Other industrial activities | 0.0000 | 1.0000 | 0.0003 | 1.0003 |

**Table A5.** Elasticity of the demand ($\varepsilon$) of water flows, Ceará.

| | Direct Water Consumption | $\Gamma_{D_j}*$ | $\Gamma_{D_i}*$ | $\Gamma_T$ |
|---|---|---|---|---|
| 0101 Agricultural | 82,674,089,052 | 0.9996 | 1.0071 | 1.0032 |
| 0201 Extractive industries | 7,921,305,183 | 1.0005 | 1.0033 | 1.0032 |
| 0301 Manufacture of food products | 7,448,717,530 | 1.0067 | 0.9994 | 1.0032 |
| 0302 Beverage manufacturing | 5,232,444,991 | 1.0000 | 0.9992 | 1.0032 |
| 0303 Manufacture of textile products, clothing and accessories, footwear, and leather goods | 4,680,501,360 | 1.0016 | 0.9991 | 1.0032 |
| 0304 Manufacture of wood products, excluding furniture, pulp, paper, and paper products and printing and reproduction services of recordings | 4,308,846,262 | 1.0008 | 0.9991 | 1.0032 |
| 0305 Refining of petroleum and coke and alcohol and other biofuels | 4,151,000,299 | 1.0050 | 0.9991 | 1.0032 |
| 0306 Manufacture of chemicals | 3,725,859,123 | 1.0037 | 0.9992 | 1.0032 |
| 0307 Manufacture of pharmaceutical and pharmaceutical products | 2,526,447,085 | 1.0043 | 0.9991 | 1.0032 |
| 0308 Manufacture of rubber and plastic products | 2,422,950,822 | 1.0066 | 0.9991 | 1.0032 |
| 0309 Manufacture of non-metallic mineral products | 1,831,771,618 | 1.0012 | 0.9993 | 1.0032 |
| 0310 Metallurgy | 1,201,903,717 | 1.0002 | 0.9992 | 1.0032 |
| 0311 Manufacture of metal products, except machinery and equipment | 1,140,693,301 | 1.0033 | 0.9991 | 1.0032 |
| 0312 Manufacture of computer equipment, electronic and optical products, machinery, appliances, and electrical materials | 1,006,556,114 | 1.0012 | 0.9991 | 1.0032 |
| 0313 Manufacture of machinery and equipment | 791,452,081 | 1.0042 | 0.9991 | 1.0032 |
| 0314 Manufacture of motor vehicles, trailers and bodywork, and other transport equipment | 592,450,880 | 1.0105 | 0.9991 | 1.0032 |
| 0315 Other industrial activities | 389,533,665 | 1.0047 | 0.9991 | 1.0032 |
| 0316 Maintenance, repair, and installation of machinery and equipment | 218,965,446 | 1.0074 | 0.9991 | 1.0032 |
| 0401 Electricity and gas, water, sewage, waste management, and decontamination activities | 190,032,220 | 1.0019 | 1.0010 | 1.0032 |
| 0501 Construction | 165,006,220 | 1.0062 | 0.9992 | 1.0032 |
| 0601 Trade and repair of motor vehicles and motorcycles | 155,296,094 | 1.0077 | 1.0032 | 1.0032 |
| 0701 Transport, storage, and mail | 130,288,371 | 1.0071 | 0.9998 | 1.0032 |
| 0801 Lodging services | 113,464,428 | 1.0011 | 0.9991 | 1.0032 |
| 0802 Food services | 42,905,922 | 1.0003 | 0.9996 | 1.0032 |
| 0901 Information and communication | 36,188,131 | 1.0005 | 0.9997 | 1.0032 |
| 1001 Financial, insurance, and related services activities | 29,809,479 | 1.0024 | 1.0001 | 1.0032 |
| 1101 Real estate activities | 13,543,610 | 0.9992 | 1.0006 | 1.0032 |
| 1201 Professional, scientific and technical, administrative, and complimentary services | 12,158,369 | 1.0020 | 1.0023 | 1.0032 |
| 1301 Administration, defence, education, and public health and social security | 7,708,586 | 1.0072 | 0.9991 | 1.0032 |
| 1401 Private education and health | 2,609,629 | 1.0012 | 1.0001 | 1.0032 |
| 1501 Arts, culture, sport and recreation and other service activities | 2,533,524 | 1.0005 | 0.9994 | 1.0032 |
| 1601 Domestic services | 1,657,177 | 1.0075 | 0.9991 | 1.0032 |

Note: * Where $i$: Lines (represents the supply of goods and services) and $j$: Columns (represents the demand for goods and services).

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
