# Peer review of "Macroeconomic Accounting of Water Resources: An Input-Output Approach to Linkage Analysis and Impact Indicators Applied to the State of Ceará, Brazil"

_water, doi:10.3390/w13060869_

Round 1

Reviewer 1 Report

The research is interesting in terms of identifying the relevant sectors for monetary and water flows.

I would suggest improving the presentation of the paper. The way it is currently presented, there a lot of long paragraphs with lots of numbers, which makes it hard for the reader to follow through. Also, consider changing the number format. 

The headings in the document are not clear, you make a jump from 3.2.4 to 4.3 -- I don't understand whether there is a section missing or a misnumbering of the section. Regardless, section 3 is very long. I would suggest keeping a results section and adding a discussion section because there is practically no discussion of what the results mean in the long term or how they would affect any water management strategy moving forward, which is, the point of the work.

Other comments:

Consider changing the number formatting: 6,8081 is sixty-eight thousand eight hundred and 81 while 6.8081 is six point eight thousand and 81.

Line 17: the objective of this work is to identify…

Line 18: , to estimate

Line 19: ,and to support

Line 20: The following methodologies were used (remove for this purpose)

Line 28: With regards to the water flows

Line 37: what is it?

Line 63: missing verb

Line 86: for this purpose,

Line 86: the justification of the use of the Rasmussen Hirschman indices is not clear. Clarify.

Line 92: water scarcity problems.

Line 105: Perhaps you meant located instead of inserted in the semi-arid region

Line 117: What is they?

Lines 125, 127: taking into consideration instead of considering

Lines 113, 131, 140: what do you mean by source author?? Please remove all over the paper

Line 141: what does “exposed” mean? Consider replacing

Lines 368, 487: Weak

Line 398: consider replacing the word exposes all over the work

Line 486: Consider adding a phrase to link between the headings

Line 550: that the induced effect

Author Response

Dear editor: We are grateful to you and the reviewers for your suggestions and for taking the time to review this manuscript. We described below the changes we have made in the manuscript, considering the suggestions dos reviewers. We highlighted all changes in yellow in the manuscript. Also, we made several changes in the text to improve the language and style (Red color in manuscript).

Please see the attachment for check a point-by-point response to the reviewer’s comments.

Reviewer 2 Report

Please, see attached file

Author Response

Dear editor:  We are grateful to you and the reviewers for your suggestions and for taking the time to review this manuscript. We described below the changes we have made in the manuscript, considering the suggestions dos reviewers. We highlighted all changes in yellow in the manuscript. Also, we made several changes in the text to improve the language and style (Red color in manuscript).

Please see the attachment for check a point-by-point response to the reviewer’s comments.

We report that the article was submitted to the special edition of Virtual Water Trade and Water Resources Economics and we received review comments requiring analysis of physical water flows distributed by water supply systems, water networks, and losses in water distribution systems. 

Reviewer 3 Report

The manuscript concerns the economic analyses  linking production sector with water resources. The paper presents the interactions between the flows of water resources and financial resources using the Input-Output Matrix method. The data collected for state of Ceara have been taken as an example for the use of the method presented by the authors in this study. It can be seen that the authors put a lot of work into writing the article, analyzing a large amount of data and interpreting it. My opinion on the manuscript is positive and contains essentially no critical remarks. As the journal is oriented towards water management specialists rather than economists, in the description of the methodology it would be advisable to describe the Rasmussen and Hirschmann indices and their role in the context of this work in a little more detail, and not just to provide references to relevant economic literature.

Author Response

(The authors gave the same response as above.)

Reviewer 4 Report

You perform a lot of theoretical and empirical work but it is presented in such a way that it is very difficult to discriminate which are your real contributions and which are re-elaborations of well-known concepts. The main reason of this may be due to the fact that you attempt to explain both the purely monetary flows and the water relationships of the economy of the region. I do think that you should focus on the second aspect (taking into account the review in which you are trying to publish).

Once focused on water-economy relationship, you could find that the main theoretical part has been extensively done many years ago in various papers (for instance, in Ecological Economics, Ecological Modelling or Water Resources Research, to name a few). The way as it is, your review of the literature in the introduction section seems random when it comes to water-economy interactions. That may be the reason why you state that there are gaps in the literature. In summary, I recommend you to follow the seminal papers on this topic and write down only your contributions to them (if any).

As an example of the above, let’s take your called ‘family’ sector in Page 16, line 512. It is difficult to distinguish if you are referring to the families as consumers of goods in general (final demand) or the families as water consumers (household water consumption). I think that if you finally focus on water and you take the well-established naming, you can avoid such confusions. Another example could be the use (or the lack of) the term ‘virtual water’. This concept refers to the total (direct or indirect) content of water embodied in a specific production, export, import or consumption. It has been used in the literature for the assessment of water trade (especially for agricultural goods) or, in general, water footprints of regions, sectors or whatever. Instead, you have adapted the term ‘field of influence’ to the water realm. It is very difficult to know what you really mean by this concept in this case.

The main proof of all I have said so far is the conclusions section 5. What do we find there? The section contains an extensive list of numerical results without any structure and three final paragraphs referring to further research in which it is suggested ‘opening of the matrix to the agricultural sector’ (which is precisely the great water consumer). If the analysis were more ‘water oriented’, as I suggest, this addition would have appeared unavoidable from the very beginning.

All those aspects I raised above lead me to recommend a major revision. I think that a very shortened and precise paper could be ‘distilled’ from this draft. In any case, many of the detailed tables should be included in annexes and provided upon request. Only the comprehensive and informative graphs and figures should be included in the paper to make easy for the reader to follow the text.

There are minor questions here and there but I think they are secondary for the major revision I suggest. For instance, there is an ‘ou seja’ in page 9 line 321. In page 11, line 399, it is very strange that real estate activities have precisely such a big forward linkage.

Summing up, the scope of the paper is perhaps too ambitious aiming at analyzing both the monetary and water structure of the regional economy. There is a bunch of seminal papers that aim specifically to the “projected effects of water use on the productive systems of the economy and the analysis of the sensitivity of economic sectors to a variation in the final demand for water resources” (a gap you point at in lines 78-80 in page 2). It would be worthwhile to apply this known methodology to the specific case at hand while making, if it is possible, some contribution to those methods.

Author Response

(The authors gave the same response as above.)

Round 2

Reviewer 2 Report

In my review I have put forward several suggestions to the manuscript and the authors made changes in the text improving the language and style.

I carefully read the authors answers  and it is clear to me that they intend to propose a purely economic analysis for the special issue. However, I think that this journal is not directed only to economists and the topic addressed in the manuscript is interdisciplinary. As physical and structural aspects related to water resource supply can alter the results obtained with the simple economic approach, this matter should be mentioned in the conclusions, but I'll leave the decision to the Editor.

Author Response

We are grateful to you and the reviewers for your suggestions and for taking the time to review this manuscript.  

Please see the attachment for check a point-by-point response to the reviewer’s comments.

Reviewer 4 Report

In my first revision I said that “very shortened and precise paper could be ‘distilled’ from this draft”. Now I realize that this advice has not been followed therefore my opinion could not change. True, there has been some improvements in the way the paper is presented (especially the results section which is now separated-extracted from the conclusions section), but the article is more or less the same. That is why my general recommendation is, more or less, the same. I will detail and precise once again my main doubts-objections.

There is no such a thing as “family sector” in the input-output literature. Thus, it is very problematic to interpret the meaning of the family sector as a key one, whether in terms of production or in terms of water. If you read some of the seminal papers on the issue you will find out that you cannot convey a key sector study with such a weak and imprecise concept. Your definition in lines 154-161 is useless in a key sector classification, whether in monetary or water use terms. As long as one of your major conclusions steams on this sector, it is very difficult to interpret what this could mean.

In your coverletter you say:

“However, this article sought to associate the Insumo-Produto model with techniques used only in the area of knowledge of Applied Economic Sciences and that had not yet been used for the area of Water Resources.”

And then you quote some recent articles that use the Input-Output techniques as seminal articles. In most of these “seminal articles” you can discover that they do quote papers in the area of water resources since no less than two decades ago. I guess that this is the reason why the very foundations of your study are so weak.

With regard to the results, let’s take the explanation of one of them. Explaining the income multiplier, you say (lines 753-756):

“This indicates that the increase of a 1m³ water unit in the final demand of this sector would lead to an increase in the sector's production of R$4,893.68 in the wage mass of workers employed in the economy.”

The final demand of this sector is different to the water that this sector demands. As a result, there is no causal link between the water demand of this sector and its production (and the wages payed), as your sentence suggests. This is but a mere example of the conceptual confusion the weak conceptual definition leads you to. Furthermore, that is also the reason why I recommended you to separate the monetary and water linkage (or at least to choose one of them). In the coverletter you answer that the other reviewers did not suggest such a thing. So, my doubts about the confusion remain. I do think that a shorter paper focused in the water content of the region production and the income elasticities of possible water shortages with respect to income and employment would be more useful and clear. Summing up, you should decide what specific aspect you want to highlight and then perform your key sector methodology to that specific purpose.

Author Response

(The authors gave the same response as above.)

Round 3

Reviewer 4 Report

The text has somehow improved and you finally followed at least one of my recommendations. I suggest you read more on this literature. The text could be useful as a first approach to water management in this particular area. In other words, I find that this more focused version could be of interest for empirical grounds. As you can imagine, my methodological concerns remain.

There are two minor issues:

Line 386 “where low water availability is low due to the combination”

It is neither necessary nor convenient, put the numbers with three decimal places, pretending an accuracy which does not appear to be the case.

Author Response

We are grateful to you and the reviewers for your suggestions and for taking the time to review this manuscript. Please see the attachment.
